# Risk Factors and Genetic Insights into Coronary Artery Disease-Related Sudden Cardiac Death: A Molecular Analysis of Forensic Investigation

**DOI:** 10.3390/ijms26083470

**Published:** 2025-04-08

**Authors:** Xiangwang He, Linfeng Li, Dianyi Zhou, Zhi Yan, Min Liu, Libing Yun

**Affiliations:** Department of Forensic Medicine, West China School of Basic Medical Sciences & Forensic Science, Sichuan University, Chengdu 610041, China

**Keywords:** coronary atherosclerosis, sudden death, whole-exome sequencing, variant, genetics, forensic medicine

## Abstract

Sudden cardiac death (SCD) is a major cause of mortality among patients with coronary artery disease (CAD). This study aimed to identify risk factors for CAD-related SCD (SCD_CAD_) through autopsy data and genetic screening with a particular emphasis on rare variants (minor allele frequency < 0.01). We included 241 SCD_CAD_ cases (mean age 54.6 ± 12.8 years, 74.7% male) verified by medico-legal examination and 241 silent CAD controls (mean age 53.6 ± 15.2 years, 25.3% female) who died from severe craniocerebral trauma. Information about death characteristics was obtained from questionnaires, police reports and autopsy data. Whole-exome sequencing was performed on myocardial tissue samples. Polygenic risk score (PRS) from a previously validated model was applied and rare variant pathogenicity was predicted using in silico tools. SCD_CAD_ victims predominantly died at night and showed higher mortality rates during summer and winter months, with more complex coronary disease. Nocturnal time (adjusted odds ratio [AOR] = 3.53, 95% CI: 2.37–5.25, *p* < 0.001), winter (AOR = 2.06, 95% CI: 1.33–3.20, *p* = 0.001), multiple vessel occlusion (AOR = 1.79, 95% CI: 1.16–2.77, *p* = 0.009), right coronary artery stenosis (AOR = 2.38, 95% CI: 1.54–3.68, *p* < 0.001) and unstable plaque (AOR = 2.17, 95% CI: 1.46–3.23, *p* < 0.001) were identified as risk factors of SCD_CAD_. The PRS score was associated with a 60% increased risk of SCD_CAD_ (OR = 1.632 per SD, 95%CI: 1.631–1.633, *p* < 0.001). Genetic analysis identified MUC19 and CGN as being associated with SCD_CAD_. We identified both hereditary and acquired risk factors that may contribute to cardiac dysfunction and precipitate SCD in CAD patients, thereby facilitating the prevention and early recognition of high-risk individuals.

## 1. Introduction

Coronary artery disease (CAD) represents one of the most significant global health threats and is the leading cause of sudden cardiac death (SCD) [1]. Due to its acute nature, SCD victims experience sudden cardiac symptoms followed by rapid death, leaving minimal time for intervention [2]. The annual incidence of SCD reaches 6 per 100,000 individuals in developed countries, with over 80 percent of death induced by CAD [3,4]. Hence, it is important to adopt more effective prevention strategies for CAD-induced SCD (SCD_CAD_).

Although SCD is the worst outcome of CAD, only a part of patients would experience this adverse event. Some patients with severe coronary atherosclerosis consistently exhibit asymptomatic conditions and maintain nearly normal cardiac function. Consequently, their CAD remains undetected until it is incidentally revealed during a medico-legal autopsy conducted for unrelated reasons. In addition, severe CAD could lead to acute myocardial ischemia and heart failure in some patients, who exhibit obvious symptoms (i.e., chest pain and breathing problems) [5]. These individuals do not succumb rapidly, thus affording them the opportunity for timely treatment. Hence, the prognosis of the above two types of CAD patients has much improved, which does not significantly contribute to the mortality of CAD. However, we observed lots of patients experienced an unexpected heart attack and even SCD. These patients, who were previously unaware before death, suffered SCD as the first presentation of CAD. Forensic autopsy revealed no lethal pathological changes in these cases, with the exception of some signs of acute myocardial ischemia, when compared with silent CAD patients and symptomatic in-hospital patients.

Genetic variations are recognized as a critical component in understanding the etiology of CAD. While numerous common genetic variants have been implicated in CAD susceptibility, large-scale studies focusing on rare variants (with allele frequencies less than 1%) have offered several distinct advantages as follows [6]: (1) identifying novel rare protein-altering variants in specific genes that are associated with disease; (2) providing supportive evidence for putative drug targets for novel therapies; and (3) uncovering new pathways and biologic mechanisms. Additionally, research on site-specific rare variants may also enhance our understanding of interplay between somatic and germline mutations, further elucidating disease mechanisms [7]. In a recent integrative analysis combining exome sequence data from three large datasets (UK Biobank, All of Us Research program and Bio*Me* Biobank), 12 rare or ultrarare (with allele frequencies less than 0.01%) variants in 11 genes were associated with a machine learning-based in silico score of CAD, which provided new insight into CAD biologic mechanisms [8]. Furthermore, a study focusing on structural variants (SVs) of CAD patients revealed that 75% of total SVs were rare with a significant proportion overlapping the USP36 gene region, suggesting its potential contribution to CAD risk [9]. Meanwhile, genetic variants, particularly harmful mutations, have been reported in numerous cases of sudden unexplained cardiac arrest, which could potentially be used to identify at-risk individuals [10]. These findings underscore the importance of exploring rare genetic variants in advancing our understanding of SCD_CAD_ pathogenesis.

Hence, we hypothesized that many unveiled internal and external triggers occurring in some CAD patients could result in SCD. In the present study, we summarized the characteristics of SCD_CAD_ patients based on autopsy data and genetic information obtained through whole-exome sequencing, which was used to explore subtle detrimental variants that could potentially contribute to SCD_CAD_. The purpose of this study was to identify both internal and external risk factors in victims of SCD_CAD_, with the aim of aiding in the prevention and early recognition of high-risk individuals.

## 2. Results

Totally, 241 SCD_CAD_ victims and 241 controls were included in this study. All victims had no history of CAD before death. The basic characteristics and autopsy findings of the study population are presented in Table 1. The mean age of SCD_CAD_ victims was 54.6 ± 12.8, with a predominant male distribution. Most of the victims died from 18:00 to 6:00 (+1 day), which was statistically different from traumatic victims (65.1% vs. 36.1%, *p* < 0.001). The mortality was higher in winter, and showed a significant difference between the two groups (34.4% vs. 22.0%, *p* = 0.002). The overwhelming majority of deaths happened in out-of-hospital locations. Through forensic examination and pathological analysis, all study populations suffered from severe coronary atherosclerosis (Figure 1A). There were 151 SCD_CAD_ victims who had at least one branch of coronary artery with almost occlusion (the vessel was narrowed more than 75 percent), compared with 104 patients in the control group (*p* < 0.001). Furthermore, we found a large number of SCD_CAD_ victims with multi-vessel (≥3 branches) coronary atherosclerosis, which had a statistical difference (33.6% vs. 23.2%, *p* = 0.012). The atherosclerosis of coronary artery was in the left anterior descending (LAD) in 190 victims (78.8%), in the right coronary artery (RCA) in 189 victims (78.4%) and in the left circumflex (LCX) in 115 victims (47.7%). The stenosis rate of RCA was significantly higher in the SCD_CAD_ group (78.4% vs. 61.0%, *p* < 0.001). The signs of unstable plaque were observed in 122 victims (50.6% vs. 36.1%, *p* = 0.001). The most frequent types were calcification (93/122) and plaque inflammation (39/122) among SCD_CAD_ victims. In the logistic regression analysis, adjusted for age and sex, nocturnal time (adjusted odds ratio [AOR] = 3.53, 95% CI: 2.37–5.25, *p* < 0.001), winter (AOR = 2.06, 95% CI: 1.33–3.20, *p* = 0.001), multiple vessel occlusion (≥3 branches) (AOR = 1.79, 95% CI: 1.16–2.77, *p* = 0.009), RCA stenosis (AOR = 2.38, 95% CI: 1.54–3.68, *p* < 0.001) and unstable plaque (AOR = 2.17, 95% CI: 1.46–3.23, *p* < 0.001) were identified as being associated with SCD_CAD_, which were considered as the risk factors (Figure 1C).

The polygenic risk score (PRS) evaluation identified 1102 matched SNPs in the SCD_CAD_ group and 1072 matched SNPs in the controls, resulting in significantly higher scores in the SCD group (Figure 2B, *p* < 0.001). The risk of SCD_CAD_ increased 60% per standard deviation (SD) of the PRS (OR = 1.632, 95%CI: 1.631–1.633, *p* < 0.001). Furthermore, 5655 deleterious variants in the SCD_CAD_ group (Figure 2A) and 3198 deleterious variants in the control group were identified by at least two in silico tools, resulting in 4248 mutant genes in the SCD_CAD_ group and 2639 mutant genes in the controls by ANNOVAR (https://annovar.openbioinformatics.org/en/latest/ (accessed on 10 July 2021)). To compare the distribution of mutant genes between the two groups, Fisher’s exact test was performed, revealing that three genes exhibited statistically significant differences. The distribution of mutant genes is depicted in Figure 2C. In summary, rare deleterious variants in the MTRNR2L5 gene were identified in 10 SCD_CAD_ patients and 1 non-SCD_CAD_ individual. Additionally, rare deleterious variants in the CGN and MUC19 genes were found in seven and nine SCD_CAD_ patients. Notably, these variants were not observed in the control group. Further analysis using Phenolyzer identified members of Mucin 19 (MUC19) and Cingulin (CGN) as being associated with SCD_CAD_, highlighting their potential role in disease pathogenesis.

In particular, two InDels in MUC19 were carried by nine CAD-related SCD victims (Table 2). The InDel rs1948650929 caused a complete alteration in the coding sequence, resulting in a loss-of-function due to the production of an abnormal polypeptide chain. The variation was extremely rare, with a frequency of 0.000015 in gnomAD, and was not detected in the ExAC or 1000 G. An additional 10 bp insertion (NC_000012.11: g.40883909_40883910insACAGGGACAACTGGACTATCAGCTGAAGCAACAGAGATAACTGGACTATCAGCT(G)4TG), identified in one SCD victim, was absent from all three genetic databases. No data were found for these two InDels in Han Chinese individuals. Additionally, four SNPs located in the CGN gene were found in seven SCD individuals. These included missense variants rs567100227, rs769053104, rs41272467 and rs931963460. Among the seven SCD individuals, three carried the rs567100227 variant, which alters alanine to valine and has specific MAFs of 0.000125 in ExAC, 0.0000521 in gnomAD and 0.0004 in 1000 G. The variant rs41272467 exhibits specific MAFs of 0.0008 in ExAC, 0.006 in gnomAD and 0.001 in the 1000 G. The variant rs769053104 was not reported in ExAC, gnomAD and 1000 G. However, with focus on the Chinese population, rs567100227 has the highest MAF of 0.004167 in Qinghai province, rs769053104 has the highest MAF of 0.001104 in Inner Mongolia and rs41272467 has the highest MAF of 0.0008 in Hebei province (Figure 2C). The variant rs931963460 was not reported in any of these databases, suggesting that further confirmation from other studies may be necessary.

## 3. Discussion

In this study, we summarized the characteristics of SCD_CAD_ victims and revealed notable features that might be associated with the risk of SCD_CAD_. The analysis of association indicated that nocturnal time, winter, multiple vessel occlusion, RCA stenosis and unstable plaque were risk factors for SCD_CAD_. Through both polygenic and monogenic approaches, the CAD PRS model was helpful in SCD_CAD_ risk stratification and SCD-related rare variants in the MUC19 and CGN genes could be considered as potential biomarkers of SCD_CAD_.

Similar to other types of SCD, the results demonstrated male patients have a higher risk of SCD_CAD_ than females within the same susceptible age interval [11]. The majority of the SCDs happened at nocturnal time when most people slept. One possible reason is obstructive sleep apnea, which had been proved to be a risk factor for SCD [12]. Additionally, staying up late could also be a trigger to increase the risk of CAD and SCD [13,14]. Notably, winter was the season with the highest incidence, which reminded CAD patients undergoing a low temperature are at greater risk of SCD. Moreover, low temperature could induce blood pressure fluctuation and an increase in myocardial oxygen consumption [15,16]. Recent studies revealed that a short-term (2–7 days) low ambient temperature was associated with both obstructive and non-obstructive CAD [17], while a decreasing temperature 27 days prior was associated with increased risk of SCD [18]. Our results showed that most SCD_CAD_ occurred out of hospitals and this part of CAD patients had little chance of medical help. So, the most effective ways of preserving life for them are prevention, early recognition and necessary interventions. We observed SCD victims had a more complex and serious coronary disease than non-SCD individuals, which might be more likely to trigger cardiac dysfunction. As a main coronary branch, RCA is very important for the blood supply of a sizeable portion of the myocardium, especially for the cardiac conduction system. Flow blockage of RCA not only leads to myocardial ischemia in the corresponding site but also results in the dysfunction of the heart’s electrical system due to disturbing the cardiac rhythm and inducing malignant arrhythmia [19]. Hence, we speculated the reason for the obstruction of RCA was associated with the risk of SCD_CAD_. The weight of the heart in both the SCDs and controls was raised compared with normal [20], but there was no significant difference between the two groups. The most common reasons for the increased heart weight are myocardial hypertrophy and cardiac fibrosis, which possibly were not risk factors for SCD_CAD_ from the findings of this study.

Genetic factors, encompassing both polygenic and monogenic components, had been established as key contributors to the incidence and progression of CAD, especially in the case of premature CAD [21]. This study extends these findings by demonstrating that genetic factors also play a significant role in SCD attributed to CAD. The PRS derived from a previously validated model exhibited significantly higher scores in the SCD_CAD_ victims compared to the controls. Additionally, the PRS was associated with a 60% increased risk of SCD_CAD_ (OR = 1.632 per SD, 95%CI: 1.631–1.633, *p* < 0.001). This finding is consistent with the results from the sex-stratified Cox regression models of CAD (HR = 1.71 per SD, 95%CI: 1.68–1.73, *p* < 0.0001) reported in the original study [22]. Collectively, these results underscore the utility of CAD-PRS in identifying individuals at risk of SCD_CAD_. Considering monogenic genetic factors, a higher proportion of CAD-related SCD victims carried the mutant MUC19 gene, which is a member of the gel-forming mucin family and mainly expresses in glandular tissues [23]. However, MUC19 could express in the epithelium under disease conditions [24] and was associated with endoplasmic reticulum (ER) stress-related inflammation in the intestinal epithelium of IBD patients [25]. The ER plays a pivotal role in maintaining cardiovascular function, and disruptions to ER homeostasis, such as ER stress, can have detrimental impacts on the health of the cardiovascular system [26]. Moreover, MUC19 was suggestively involved in lipid synthesis and associated with circulating omega-3 polyunsaturated fatty acids [27]. CGN was identified by our analysis as another implicated gene in the pathophysiology of SCD_CAD_. CGN is a key component of tight junctions in human epithelial and endothelial cells, playing a crucial role in preserving the integrity of these cell barriers during both physiological and pathological processes [28]. Endothelial barrier function was decreased in CGN knockout mice as shown by a previous study [29]. The increased permeability of the endothelium, followed by the subsequent accumulation of low-density lipoproteins (LDLs) in the intima, contributes to the progression of atherosclerosis [30]. Furthermore, our study highlights that genetic variants in the MUC19 and CGN gene regions are more frequently observed in Han Chinese individuals compared to global datasets. Notably, one specific variant in CGN, rs567100227, has the highest MAF of 0.004167 in Qinghai, a region adjacent to Sichuan where our study was conducted. In summary, our findings provided evidence that both polygenic and monogenic approaches may offer valuable insights for the early screening of SCD_CAD_.

The evaluation of SCD_CAD_-related risk factors is beneficial for the early recognition of high-risk people and the prevention of SCD. The identified individuals could pay more attention to their cardiovascular status. Even for patients who already have CAD, SCD could still be prevented by limiting exposure to some avoidable risk factors and beginning cardiac invasive treatment if necessary. In addition, both genetical and morphological risk factors would help the forensic investigation of SCD, which would facilitate a more accurate pathological diagnosis and a better explanation of the mechanism of SCD_CAD_.

However, some limitations were inevitable in this study. Firstly, the size of the samples was a major limitation. Due to the influence of weakening the serviceability of the results, it needs to be understood with caution. The effects of the identified risk factors should be verified in further larger studies. Moreover, insufficient clinical data from SCD victims probably resulted in some potential clinical risk factors being unverified in the study population.

## 4. Materials and Methods

### 4.1. Study Design and Ethical Approval

The present study was a retrospective analysis of case–control data. This study was approved by the ethics committee of Sichuan University (ID: K2022003). Written informed consent was clarified by the families of every participant.

### 4.2. Study Setting and Samples Collection

We enrolled victims with complete post-mortem examinations from the Forensic Medical Identification Center of West China Sichuan between 2010 and 2021. The SCD_CAD_ group was defined by the following criteria: (1) the witnessed death happened within 1 h of the acute heart attack, or the unwitnessed death happened within 24 h of the patient last being seen apparently healthy, and (2) the death was entirely attributable to CAD. An equal number of individuals who died from severe craniocerebral trauma with severe silent CAD (a coronary stenosis level greater than 50%) were assigned to the control group (Figure 1A). These controls were matched with the study group according to gender, age (±5 years) and the year of death (±3 years). In cases where multiple candidates met the criteria, a random selection method was employed to choose the control. Cases for which a suitable match could not be found were excluded from the study. Sociodemographic characteristics and information of death (age, time of death, season of death and place of death) were obtained from questionnaires of victims’ dependents and police reports. Due to the sudden and unexpected nature of deaths in the study population, the lack of prior specific medical examinations means that we were unable to obtain the clinical characteristics of the deceased individuals. Of the 241 SCD_CAD_ group and 241 controls used in this study, 30 cases and 30 controls were previously analyzed and published in an earlier study [11]. These initial analyses provided foundational insights for the current comprehensive investigation.

A complete medico-legal examination, including autopsy, histological observation, toxicological analysis and other required laboratory and molecular tests, was performed by senior forensic pathologists until the cause of death was clear. Coronary atherosclerosis was diagnosed by macroscopic and microscopic inspection, and the degree of stenosis was estimated using digital pathological section. During autopsies, the whole heart with several myocardial and coronary samples was inspected carefully. Left anterior descending (LAD), left circumflex (LCX) and right coronary artery (RCA) were separated for inspection. Especially, we defined unstable plaque based on histopathological examination using hematoxylin and eosin (H&E) staining when any of the following conditions were met: (1) calcification: indicated by distinct calcium deposits within plaque; (2) plaque inflammation: characterized by the presence and density of macrophages and lymphocytes; (3) intraplaque hemorrhage: identified by erythrocytes within the plaque; (4) erosions: evidenced by endothelial discontinuities and thrombus formation. Fatal CAD, which could cause sudden death, was considered when the degree of stenosis was greater than 50%, and any of the following pathological changes was matched: (1) complicated plaques, (2) coronary thrombosis, (3) acute myolysis, (4) myocardial hemorrhage and (5) reported cardiac symptoms within 1 h of death [31]. Approximately 10 g of left ventricular septal myocardial tissue was collected, fixed in formalin and then embedded in paraffin for preservation and whole-exome sequencing.

### 4.3. Whole-Exome Sequencing and Genotyping

To ensure sample quality, we selected specimens from 40 unrelated SCD_CAD_ victims and 23 controls for whole-exome sequencing. All corpses were examined or stored at temperatures below 20 °C within 1 h after death, and tissue samples were stored for no more than 2 years. Genomic DNA was isolated from collected myocardial tissues by the black PREP FFPE DNA Kit (Analytik Jena AG, Jena, Germany), by following the manufacturer’s protocols. Exome capture was performed with AIExomeV2 Enrichment Kit (iGeneTech, Beijing, China). The construction and quality testing of the DNA library was completed by a commercial service. High-throughput sequencing was carried out on Illumina systems (Illumina, San Diego, CA, USA) according to the manufacturer’s instructions.

Trimmomatic was used for the quality assessment of sequencing data [32]. The reads were removed if sequencing depth < 20×, genotype quality < 30 or frequency deviation > 0.2. Clean reads were aligned to the reference genomes (UCSC hg19) in the Burrows–Wheeler Alignment Tool, and the results were saved in BAM format [33]. SAMtools and Genome Analysis Toolkit were used to search single-nucleotide variants (SNVs) and insertion–deletions (Indels) in BAM files [34,35].

### 4.4. Evaluation of Polygenic Risk Score

We applied the polygenic risk score (PRS) model of CAD developed by a previous study to evaluate the difference between SCD_CAD_ victims and controls [22]. The meta-score, known as metaGRS, was constructed by integrating three genetic risk scores (GRS46K [36], FDR202 [37], 1000 Genomes [37]), encompassing a total of 1.7 million genetic variants. Notably, metaGRS included genetic variants with a minor allele frequency (MAF) greater than 0.001, enabling the assessment of the effects of rare genetic variants (MAF < 0.01). We conducted the PRS evaluation of genetic variants passing the Hardy–Weinberg equilibrium test threshold (<0.05) using PLINK [38]. We conducted a Z-score transformation of the PRS and employed a logistic regression model to assess the association between PRS and the risk of SCD_CAD_. This approach allowed us to evaluate the per SD effect of the PRS on the risk of SCD_CAD_.

### 4.5. Prediction and Annotation of Variant Pathogenicity

We used Sorting Intolerant From Tolerant (SIFT) [39], Polymorphism Phenotyping v2 (PolyPhen-2) HIDV [40], PolyPhen-2 HVAR [40], Likelihood Ratio Test [41], MutationTaster2 [42], MutationAssessor [43], Combined Annotation-Dependent Depletion (CADD) [44], Functional Analysis through Hidden Markov Models (FATHMMs) [45] and FATHMM-MKL [46] to predict the pathogenicity of detected rare variants. Given our focus on rare variants, the minor allele frequency (MAF) threshold for potential damaging mutations was set at less than 0.01. Variants were considered deleterious if they were predicted to be deleterious or harmful by at least two predicted algorithms mentioned above. To investigate the global MAF, we accessed three public genetic databases: Genome Aggregation Database (gnomAD), Exome Aggregation Consortium (ExAC) and 1000 Genomes reference data (1000 G). To account for ancestry differences, we further retrieved the MAF of identified variants from PGG.Han, which archives 324,214,115 genome-wide single nucleotide variants of 137,012 Han Chinese individuals [47]. The annotation of mutations was performed by ANNOVAR [48]. To assess the difference in the distribution of the mutant genes between SCD_CAD_ victims and controls, Fisher’s exact test was utilized. The relationship between identified mutant genes and SCD_CAD_ was further evaluated by Phenolyzer [49], which could indicate the association of candidate genes and targeted diseases with a reliable performance. The term ‘sudden cardiac death’ was the key word for the correlation analysis. The genes with a Phenolyzer score ≥ 0.01 were perceived as contributing to SCD_CAD_.

### 4.6. Statistical Analysis

Continuous variables were first assessed for normality using the Shapiro–Wilk test. Given that these variables did not fulfill parametric assumptions, comparisons between groups were performed using the nonparametric Mann–Whitney U test. Categorical variables were analyzed using the chi-square test or Fisher’s exact test, as appropriate. Multiple comparisons were corrected using the Bonferroni method. Multivariate logistic regression analysis was conducted to identify independent risk factors associated with SCD_CAD_, and results were expressed as adjusted odds ratios (AORs) with corresponding 95% confidence intervals (CIs). Statistical significance was considered when the *p*-value was less than 0.05. All statistical analyses were performed in IBM SPSS Statistics 26.0 (IBM Corp, Armonk, NY, USA).

## 5. Conclusions

In conclusion, we identified several distinctive features of SCD caused by CAD. Multiple environmental and pathological changes were associated with SCD_CAD_, including nocturnal time, winter, multiple vessel occlusion, RCA stenosis and unstable plaque, which could be considered as acquired risk factors of SCD_CAD_. From a polygenic perspective, the CAD-PRS demonstrated significant utility in risk stratification for SCD_CAD_. Furthermore, a considerable proportion of SCD_CAD_ victims had detrimental rare variants in MUC19 and CGN, which were discovered to be SCD-related genes. The genetic variants were found to be more prevalent in Han Chinese individuals. Collectively, our findings provide critical insights into the early screening and preventive strategies for SCD_CAD_, highlighting the importance of both environmental and genetic factors in risk assessment.

## Figures and Tables

**Figure 1 ijms-26-03470-f001:**
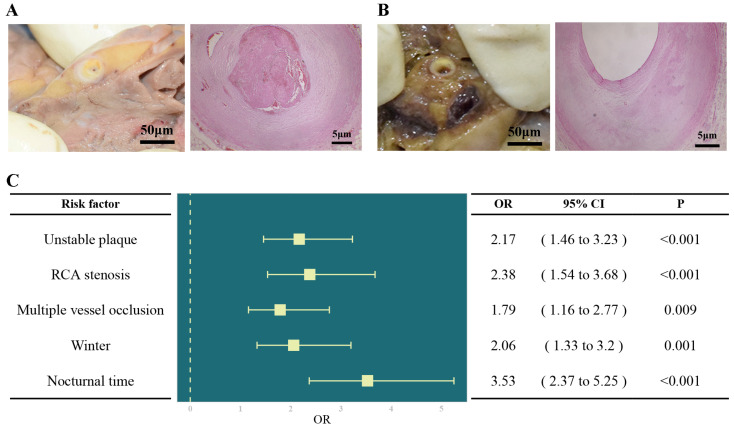
(**A**) Atherosclerotic plaques caused significant luminal stenosis in the SCD_CAD_ group through gross investigation and histological evaluation; (**B**) atherosclerotic plaques in the controls through gross investigation and histological evaluation; (**C**) risk factors related to SCD_CAD_.

**Figure 2 ijms-26-03470-f002:**
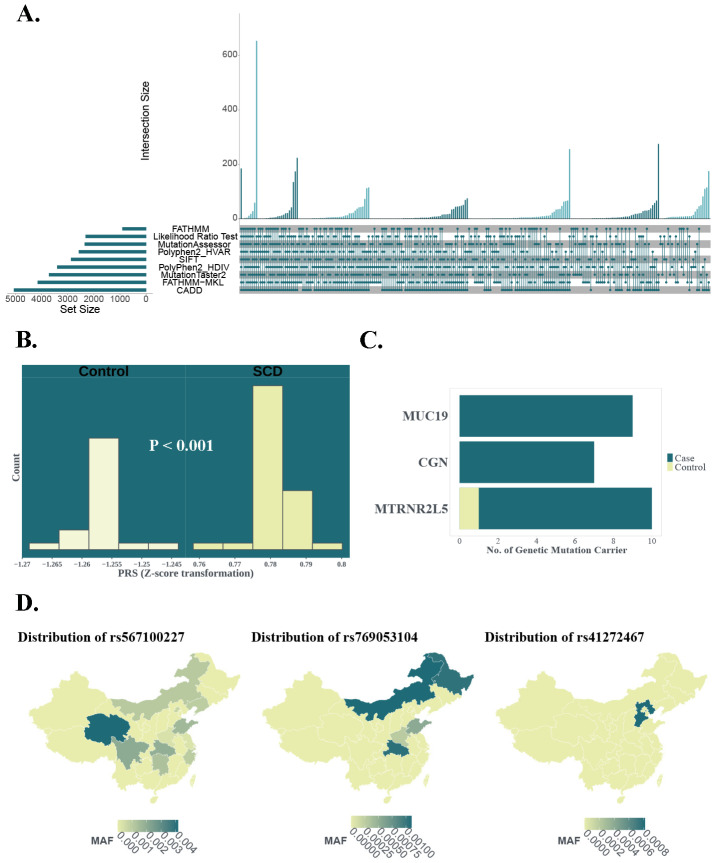
(**A**) Deleterious genetic variants identified by nine in silico tools in SCD_CAD_ group; (**B**) PRS distribution: bars indicate sample counts and *p* value from Mann–Whitney U test; (**C**) mutant genes identified by Fisher’s exact test and their distribution; (**D**) the distribution of genetic variants rs567100227, rs769053104 and rs41272467 in Han Chinese individuals.

**Table 1 ijms-26-03470-t001:** Epidemiological features of SCD patients with CAD.

	SCD_CAD_	Controls	*p*
Age (years)	54.6 ± 12.8	53.6 ± 15.2	>0.05
Gender (N/%)			>0.05
Male	180 (74.7)	180 (74.7)	
Female	61 (25.3)	61 (25.3)	
Time of death (N/%)			<0.001
Diurnal	84 (34.9)	154 (63.9)	
Nocturnal	157 (65.1)	87 (36.1)	
Season of death (N/%)			0.022
Spring	48 (19.9)	60 (24.9)	
Summer	65 (27.0)	71 (29.4)	
Autumn	45 (18.7)	57 (23.7)	
Winter	83 (34.4)	53 (22.0)	
Place of death (N/%)			
In-hospital	21 (8.7)	53 (22.0)	<0.001
Out-of-hospital	220 (91.3)	188 (78.0)	
Heart weight	407.6 ± 138.4	448.5 ± 144.5	>0.05
Number of narrowing vessels (N/%)			0.035
1	69 (28.6)	74 (30.7)	
2	91 (37.8)	111 (46.1)	
3	81 (33.6)	56 (23.2)	
Location of stenosis (N/%)			/
LAD	190 (78.8)	189 (78.4)	
RCA	189 (78.4)	147 (61.0)	
LCX	115 (47.7)	128 (53.1)	
Unstable plaque (N/%)	122 (50.6)	87 (36.1)	0.001
Calcification	93	82	
Plaque inflammation	39	11	
Intraplaque hemorrhage	15	5	
Erosion	11	2	
Others	5	0	

LAD: left anterior descending; RCA: right coronary artery; LCX: left circumflex.

**Table 2 ijms-26-03470-t002:** Variants in significant mutant genes carried by SCD_CAD_ victims.

Gene	Variant	ID of SNP	Protein	Functional Annotation	ACMG	No. of SCD_CAD_
MUC19	NC_000012.11: g.40884241_40884242ins [54 bp] (GRCh37.p13)	rs1948650929	/	Frameshift insertion	VUS	8
	NC_000012.11: g.40883909_40883910ins [64 bp] (GRCh37.p13)	/	/	Stop gained and frameshift insertion	VUS	1
CGN	c.2000C>T	rs567100227	p.Ala667Val	Missense	VUS	3
	c.2875G>A	rs769053104	p.Asp959Asn	Missense	VUS	1
	c.2818G>A	rs41272467	p.Ala940Thr	Missense	VUS	2
	c.1978C>T	rs931963460	p.Arg660Trp	Missense	VUS	1

Alleles of [54 bp]: GGGACAACTGGACCATCA(GCT)2GTGACTGGGTCAGCTGGACTATCAGCT(G)4TGACC; alleles of [64 bp]: ACAGGGACAACTGGACTATCAGCTGAAGCAACAGAGATAACTGGACTATCAGCT(G)4TG; ACMG, classification of variants from American College of Medical Genetics and Genomics; VUS: variants of uncertain significance.

## Data Availability

The data of this study are available from the corresponding author (Libing Yun) upon reasonable request.

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
