# Peer review of "Risk Factors and Genetic Insights into Coronary Artery Disease-Related Sudden Cardiac Death: A Molecular Analysis of Forensic Investigation"

_ijms, 2025, doi:10.3390/ijms26083470_

Round 1
Reviewer 1 Report
Comments and Suggestions for Authors
The manuscript entitled “Risk Factors and Genetic Insights into Coronary Artery Disease-Related Sudden Cardiac Death: A Molecular Analysis of Forensic Investigation” was evaluated. The work presents relevant information for the scientific community; however, some adjustments need to be made so that it can be published in this journal.
Below are some notes on possible adjustments. I kindly ask that you read it carefully and mark in red or highlight in yellow the adjustments made to the manuscript.
Abstract:
1- In the abstract methodology, please provide more information about the victims. Average age, family history and body composition, for example.
2- I believe that the results failed to explore possible correlations between genetic factors. The information about “MUC19 and 21 CGN” was very vague and needs to be further explored, since this is the theme of the study.
3- In line 20, what does the acronym “RCA” mean?
4- Regarding the conclusion, it is also necessary to conclude the proposed objectives more precisely.
Introduction:
5- In the introduction, present possible genetic factors responsible for SCD in patients with or without CAD.
6- In line 57, the acronym “SCDCAD” is in all capital letters, but in the rest of the text it is not. Please standardize it.
7- Standardize the objective described at the end of the introduction with that presented in the abstract.
Materials and Methods:
8- Provide more information between lines 62 and 64 about the ethical approval, such as the protocol number.
9- As already mentioned, the acronym “SCDCAD” presented in line 67 should be standardized in the text.
10- Adjust the formatting between lines 76 and 77.
11- In the methodology, I noticed that sociodemographic information about the victims was missing, as well as body composition, to determine whether age and weight could also be cofactors for deaths. Please describe how this information was collected and how.
12- I was unable to see in the methodology whether you performed a sample calculation to arrive at the number of victims needed to obtain adequate statistical power.
13- Describe the statistical analyses in more detail. For example, were the data subjected to normality and homogeneity tests? If so, which ones? Please indicate which nonparametric data were analyzed using statistics corresponding to the parametric ones and cite the name of the tests.
Results
14- In line 145, I did not understand what “6:00+1” means.
15- Table 1 does not show the scales for the variables. Please include them. Example: age (years). Take advantage and include the abbreviation SCDCAD and the statistical test that was used in the table caption. In addition, the information in Table 1 appears after the information in Figure 1 in the text. Therefore, you should adjust the order in which they are presented in the manuscript.
16- Please improve the quality of Figure 1. In the case of 1A, the scale was not added to the images and the caption did not detail what each photo represents. In 1B and 1C, the resolution is low, and the letters are very small. One suggestion would be to separate the figures to improve the material presented.
17- In Table 2, you need to improve the caption, including all the abbreviations.
Discussion
18- In the discussion, I felt that more information was missing on how this study could contribute clinically to patients who are considered victims of sudden death. Please provide more information on this aspect.
19- It would also be interesting for you to explain more clearly the reason for specifically studying the MUC19 and CGN genes for this study.
Conclusions
20- They tend to direct the conclusion towards the objectives.
References
21- Regarding the theoretical framework, it was found that there are many references that are more than 5 years old. Please update the framework with information from 2020 to 2025.
Author Response
Dear Reviewer,
We sincerely appreciate your thorough and insightful feedback on our manuscript. Please accept our deep apologies for the oversights in the initial submission. We have diligently addressed all formatting and linguistic concerns raised in your review, with substantive revisions marked in red text throughout the manuscript. Additionally, we have updated citations older than five years as requested (marked in red), ensuring current scholarly context.
Following editorial guidance, we have restructured the article sequence to: Introduction → Results → Discussion → Materials and Methods → Conclusion. This reorganization aims to enhance logical flow while maintaining methodological transparency, and we trust it will facilitate your review process.
Below, we firstly provide comprehensive responses to your key concerns:
- Sociodemographic Information Collection: We have comprehensively revised the methodology description (Lines 250-252) to clarify data acquisition protocols. Regarding body composition and weight metrics: Postmortem weight data were excluded due to inherent limitations in retrospective case-control studies, including frequent missing entries and potential inaccuracies from postmortem changes. Body composition analysis is not routinely prioritized in forensic pathology practice for non-metabolic death investigations. We fully acknowledge the scientific value of these parameters and will systematically incorporate anthropometric measurements into our future research protocols.
- Enhanced Genetic Correlation Analysis: Regarding the gene analysis section, in the revised version, we have improved the original analysis and added two additional analyses. We hope the results will be convincing to you.
- We have introduced rigorous statistical validation of our initial findings through Fisher's exact test (Lines 320-321). This analysis confirms the significant differential expression patterns between case and control groups, as subsequently validated by Phenolyzer prioritization (Lines 113-122; Figure 2C). The concordance between these approaches strengthens the biological plausibility of our candidate genes.
- To address clinical translation concerns, we applied polygenic risk score (PRS) model of CAD developed by previous study (Lines 294-305). Our analysis reveals compelling discriminatory capacity for SCDCAD risk stratification (Lines 107-110; Figure 2B). This represents a potential screening tool for primary prevention in clinical settings.
- Leveraging the PPG.Han database (137,012 Han Chinese individuals), we conducted focused analysis of rare variants in MUC19 and CGN genes (Lines 317-319). The MAF distribution demonstrated higher prevalence in Han Chinese compared to global datasets (Lines 137-141; Figure 2D).
- Sample Calculation and Statistical Analyses: We sincerely appreciate this insightful comment regarding statistical power considerations. The retrospective nature of our case-control study inherently limited our ability to perform a priori sample size calculations. In our study, the sample size was determined based on the availability of data and the specific objectives of the research. However, we recognize that a formal sample size calculation would have strengthened the statistical rigor of our analysis. To address this limitation, we will consider incorporating power analysis in future studies. Furthermore, we have revised the description of the statistical analysis in the methods section based on your suggestion. We sincerely apologize again for our oversight.
The following content in the table is our point-by-point revision directly to address the reviewer's insightful comment:
Part |
Comment |
Revision |
Abstract |
1- In the abstract methodology, please provide more information about the victims. Average age, family history and body composition, for example. |
Lines 13-14 |
2- I believe that the results failed to explore possible correlations between genetic factors. The information about “MUC19 and 21 CGN” was very vague and needs to be further explored, since this is the theme of the study. |
Lines 17-18; Lines 24-25 |
|
3- In line 20, what does the acronym “RCA” mean? |
Lines 22 |
|
4- Regarding the conclusion, it is also necessary to conclude the proposed objectives more precisely. |
Lines 26-28 |
|
Introduction |
5- In the introduction, present possible genetic factors responsible for SCD in patients with or without CAD. |
Lines 54-69 |
6- In line 57, the acronym “SCDCAD” is in all capital letters, but in the rest of the text it is not. Please standardize it. |
Lines 78 |
|
7- Standardize the objective described at the end of the introduction with that presented in the abstract. |
Lines 71-73 |
|
Results |
14- In line 145, I did not understand what “6:00+1” means. |
Lines 85-86 |
15- Table 1 does not show the scales for the variables. Please include them. Example: age (years). Take advantage and include the abbreviation SCDCAD and the statistical test that was used in the table caption. In addition, the information in Table 1 appears after the information in Figure 1 in the text. Therefore, you should adjust the order in which they are presented in the manuscript. |
Table 1; Lines 83-84 |
|
16- Please improve the quality of Figure 1. In the case of 1A, the scale was not added to the images and the caption did not detail what each photo represents. In 1B and 1C, the resolution is low, and the letters are very small. One suggestion would be to separate the figures to improve the material presented. |
Figure 1; Figure 2 |
|
17- In Table 2, you need to improve the caption, including all the abbreviations. |
Table 2 |
|
Discussion |
18- In the discussion, I felt that more information was missing on how this study could contribute clinically to patients who are considered victims of sudden death. Please provide more information on this aspect. |
Lines 191-200 |
19- It would also be interesting for you to explain more clearly the reason for specifically studying the MUC19 and CGN genes for this study. |
Lines 216-221 |
|
Materials and Methods |
8- Provide more information between lines 62 and 64 about the ethical approval, such as the protocol number. |
Lines 237 |
9- As already mentioned, the acronym “SCDCAD” presented in line 67 should be standardized in the text. |
Lines 241 |
|
10- Adjust the formatting between lines 76 and 77. |
Lines 249-250 |
|
11- In the methodology, I noticed that sociodemographic information about the victims was missing, as well as body composition, to determine whether age and weight could also be cofactors for deaths. Please describe how this information was collected and how. |
Lines 250-252 |
|
12- I was unable to see in the methodology whether you performed a sample calculation to arrive at the number of victims needed to obtain adequate statistical power. |
Please refer to the responses above. |
|
13- Describe the statistical analyses in more detail. For example, were the data subjected to normality and homogeneity tests? If so, which ones? Please indicate which nonparametric data were analyzed using statistics corresponding to the parametric ones and cite the name of the tests. |
Lines 328-335 |
|
Conclusion |
20- They tend to direct the conclusion towards the objectives |
Lines 346-348 |
Reference |
21- Regarding the theoretical framework, it was found that there are many references that are more than 5 years old. Please update the framework with information from 2020 to 2025. |
Lines 370-373; Lines 396-398; Lines 405-408; |
We remain deeply grateful for your expertise in strengthening this work. Should any aspects require further clarification, we would be pleased to provide additional details. Thank you for your understanding and guidance.

Reviewer 2 Report
Comments and Suggestions for Authors
This study presents a well-designed approach with interesting observations. The clinical findings align with previous studies and reinforce established knowledge. However, I have significant concerns regarding the genetic findings. CAD is a polygenic disease influenced by multiple variants, which together contribute to disease development through polygenic risk scores (PRS). It would be valuable to compare the PRS between the two groups to better understand the genetic contribution.
Major Concerns:
- Comparison of Mutated Genes:
The study reports 4,248 mutated genes in the SCDCAD group and 2,639 in the control group. However, only MUC19 and CGN were identified as associated with SCDCAD, while no target genes were reported in the control group. Was there truly no identified target gene in the control group? A direct comparison of differentially expressed genes between the two groups is needed to clarify the differences. - Cardiac-Specific Gene Analysis & PRS:
- When focusing on cardiac-related genes known to be expressed in the heart, were there any significant differences between the two groups?
- How do these findings correlate with polygenic risk scores (PRS) for CAD? Given that SNP-based PRS has been extensively studied in CAD, it would be valuable to analyze known SNPs within this dataset. Moreover, 4 SNPs were found on the SCDcad group. Again, how many were found in the control group?
- I recommend referencing studies such as Genomic Risk Prediction of Coronary Artery Disease in 480,000 Adults: Implications for Primary Prevention - PubMed to provide context and support your findings.
- Minor Allele Frequency (MAF) Cutoff Selection:
- The study sets the minor allele frequency (MAF) cutoff at 0.01 (1%). Could you clarify the rationale behind this threshold?
Minor Concerns:
- Abstract: Please provide the full term before using the abbreviation "RCA."
- Line 181: Sentences should not begin with a numeral. Consider rewording for clarity.
Final Thoughts:
This study has great potential and could make a valuable contribution to cardiology if the genetic analysis is expanded. The patient selection was strong, and refining the genetic cardiology perspective could significantly enhance the impact of this work.
Author Response
Dear Reviewer,
We sincerely appreciate your recognition of this study's significance and your thoughtful guidance in expanding the methodological framework. We have diligently addressed all formatting and linguistic concerns raised in your review, with substantive revisions marked in red text throughout the manuscript. In response to your insightful comments, we have implemented the following substantial enhancements:
(Following editorial guidance, we have restructured the article sequence to: Introduction → Results → Discussion → Materials and Methods → Conclusion. This reorganization aims to enhance logical flow while maintaining methodological transparency, and we trust it will facilitate your review process.)
- [Comment 1]: Comparison of Mutated Genes: The study reports 4,248 mutated genes in the SCDCAD group and 2,639 in the control group. However, only MUC19 and CGN were identified as associated with SCDCAD, while no target genes were reported in the control group. Was there truly no identified target gene in the control group? A direct comparison of differentially expressed genes between the two groups is needed to clarify the differences.
- [Response]: We have introduced rigorous statistical validation of our initial findings through Fisher's exact test (Lines 320-321). This analysis confirms the significant differential expression patterns between case and control groups, as subsequently validated by Phenolyzer prioritization (Lines 113-119; Figure 2C). The concordance between these approaches strengthens the biological plausibility of our candidate genes.
- [Comment 2]: Cardiac-Specific Gene Analysis & PRS: 1) When focusing on cardiac-related genes known to be expressed in the heart, were there any significant differences between the two groups? 2) How do these findings correlate with polygenic risk scores (PRS) for CAD? Given that SNP-based PRS has been extensively studied in CAD, it would be valuable to analyze known SNPs within this dataset. Moreover, 4 SNPs were found on the SCDcad group. Again, how many were found in the control group?3) I recommend referencing studies such as Genomic Risk Prediction of Coronary Artery Disease in 480,000 Adults: Implications for Primary Prevention - PubMed to provide context and support your findings.
- [Response]: According to your insightful recommendation, we applied polygenic risk score (PRS) model of CAD developed by previous study (Lines 294-305). Our analysis reveals compelling discriminatory capacity for SCDCAD risk stratification (Lines 107-110; Figure 2B). Notably, the PRS was associated with 60% increased risk of SCDCAD (OR = 1.632 per SD, 95%CI: 1.631-1.633, P < 0.001). This finding is consistent with result from sex-stratified Cox regression models of CAD (HR = 1.71 per SD, 95%CI: 1.68-1.73, P < 0.0001) reported in original study. This represents a potential screening tool for primary prevention in clinical settings.
- [Comment 3]: Minor Allele Frequency (MAF) Cutoff Selection: The study sets the minor allele frequency (MAF) cutoff at 0.01 (1%). Could you clarify the rationale behind this threshold?
-
- [Response]: We sincerely apologize for not clearly articulating the research content and objectives of the paper in the initial submission. In the introduction section, we have elaborated on rare variants (with MAF less than 1%) and their research progress in the context of CAD (Lines 54–69). Additionally, we integrated the analysis results of the PRS model and discussed the value distribution of polygenic and monogenic approaches in this study in the discussion section (Lines 191-201). Furthermore, we conducted focused analysis of rare variants in MUC19 and CGN genes by leveraging the PPG.Han database (137,012 Han Chinese individuals) (Lines 317-319). The MAF distribution demonstrated higher prevalence in Han Chinese compared to global datasets (Lines 137-141; Figure 2D).
The following content in the table is our point-by-point revision directly to address the reviewer's insightful comment:
Part |
Comment |
Revision |
Abstract |
Abstract: Please provide the full term before using the abbreviation "RCA." |
Lines 22 |
Results |
Line 181: Sentences should not begin with a numeral. Consider rewording for clarity. |
Lines 127-130 |
We sincerely thank you for your careful and thorough review, as well as the insightful suggestions you provided based on your in-depth thinking and valuable research experience. Should any aspects require further clarification, we would be pleased to provide additional details. Looking forward to hearing from you.

Round 2
Reviewer 1 Report
Comments and Suggestions for Authors
Dear authors and editor,
I would like to thank the authors for their care in adjusting the manuscript according to the guidelines outlined by the reviewers. After reading the adjusted manuscript, I suggest that it be approved for publication.
Reviewer 2 Report
Comments and Suggestions for Authors
No more comments. The authors responded very well to my questions.